# Don't Throw Away Data: Better Sequence Knowledge Distillation

## Abstract

A critical component in knowledge distillation is the means of coupling the teacher and student. The predominant *sequence knowledge distillation* method involves supervised learning of the student against teacher-decoded outputs, and is exemplified by the current state of the art, which incorporates minimum Bayes risk (MBR) decoding. In this paper we seek to integrate MBR more tightly in distillation training, specifically by using several high scoring MBR translations, rather than a single selected sequence, thus capturing a rich diversity of teacher outputs. Our experiments on English to German and English to Japanese translation show consistent improvements over strong baseline methods for both tasks and with varying model sizes. Additionally, we conduct a detailed analysis focusing on data efficiency and capacity curse aspects to elucidate MBR-$n$ and explore its further potential.

## 1 Introduction

Large language models (LLM) have shown remarkable capabilities in multilingual language understanding and translation. With careful prompting, LLMs can produce high quality translations for a range of translation languages, rivaling or exceeding that of the traditional encoder-decoder architectures translation systems (Anil et al., 2023). However, despite their superior performance, LLMs are substantially bigger, more resource-intensive, and slower than the encoder-decoder translation systems. This raises the question of how the advanced translation capabilities of the LLM can be transferred to cheaper and more efficient models that can be deployed more widely, and with a lower carbon footprint. Knowledge Distillation (KD) presents a practical solution to this issue, such that the translation outputs of a complex LLM teacher can be used to train a simpler student model. This builds on a history of distillation research, starting with Hinton et al. (2015), and extending to sequential generation models (Kim & Rush, 2016). Despite many intervening years, Kim & Rush's SeqKD approach is still widely used in translation research (Tan et al., 2019; Wang et al., 2021; Zhang et al., 2023b), specifically in its simplest version which trains the student using standard cross-entropy loss on translation sentences generated by the teacher. Recently, this black-box approach has also gained widespread use in LLM knowledge distillation (Peng et al., 2023; Zhou et al., 2023; Finkelstein et al., 2023), as many proprietary LLMs only offer APIs for user interaction.

A key question in distillation is what information from the teacher will best inform the learning of the student. While most SeqKD approaches have used greedy or beam decoding to generate teacher samples, Finkelstein et al. (2023) proposed the use of minimum Bayes risk (MBR) decoding. This decoding method generates a large pool of candidate samples, which are then compared in a pair-wise manner using a reference-based evaluation metric, and the most central sample is selected. This results in an improvement in translation accuracy over beam search decoding, as well as better student performance when used as a distillation target.

In this paper we explore ways to use deeper information from the MBR computation for better learning of student models. In particular, we show that presenting several candidates to students instead of the single sequence, results in better distilled student output. Our experiment results over two language pairs and with varying sizes of teacher and student models show that providing several sequences provides consistent improvements over 1-best MBR supervision, and generally observe increasing student performance with increasing number of supervision sequences. Overall this work

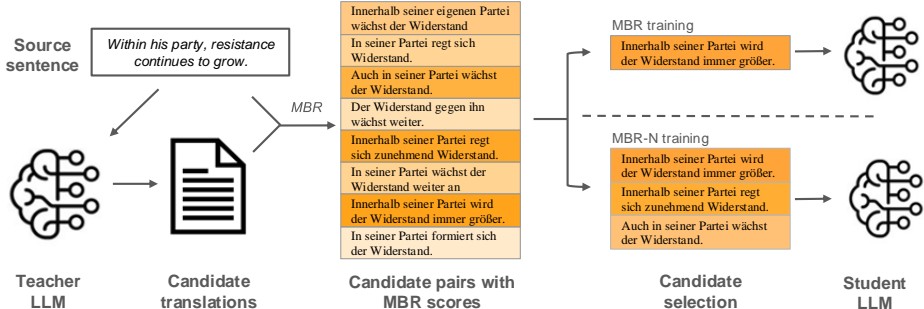

Figure 1: The workflow of MBR-$n$, in comparison to MBR, is illustrated using an example of translating a sentence from English to German with $n$=3. The steps include: a) Generating several translations from the Teacher LLM; b) Computing Bayes Risk scores (depicted by colour intensity); c) Distillation training by supervising the student with MBR output for the given source, or supervised by each of the top-3 scoring translations. Input example from WMT09-19 (en-de), with candidate translations generated by the L model.

argues against using a single point estimate for knowledge distillation, showing benefits from deeper integration between student and teacher.

Our contributions are as follows:

- We propose MBR-$n$, a method based on using $N$ candidates from MBR decoding, to better enable the student to learn to match the distribution of high-quality teacher outputs.

- We conduct extensive experiments on two translation tasks, en-de and en-ja, with leading Palm 2 models at various sizes, and show MRB-$n$ improves student performance over competitive benchmark methods.

- We conduct extensive analysis, showing our method leads to consistent improvements in data efficiency, investigate uncertainty and output diversity, and confirm the capacity gap curse is an open issue, whereby distillation's effectiveness diminishes as the teacher capacity grows substantially beyond that of the student.

## 2 METHOD

**Sequence-level knowledge distillation** (SeqKD) is a popular distillation method for translation (Kim & Rush, 2016). It works by training the student to match the sequence outputs of the teacher, commonly formulated as minimizing

$$\mathcal{L}_{\text{SeqKD}} \approx -\sum_{h \in \mathcal{H}} \mathbb{1}\{h = \hat{y}\} \log p(t = h|s)$$
$$= -\log p(t = \hat{y}|s) \tag{1}$$

where $\mathcal{H}$ is the set of all possible sequences, $p(t|s)$ is the student model, and $\hat{y}$ is the output generated by teacher model, e.g., via beam search decoding. This approach is motivated by mimicking the full distribution of the teacher, however the intractability of considering infinitely many sequences necessitates approximation, here by approximating the teacher's full distribution by its mode.

Intuitively, a better approximation of the distribution, as expounded in Kim & Rush (2016), is to generate the $n$-best translations from the teacher model for each source sentence. Their experiments indicate that this approach is not as effective for distillation as using beam search. One possible reason for this is the low diversity in beam search outputs, and thus small settings of $n$ do not expose the student to meaningful variation. In this paper we propose instead to use MBR decoding to generate $n$-best candidates, based on selecting the top MBR scoring candidates from a diverse pool of sampled outputs.

**MBR** (Kumar & Byrne, 2004) is based on the principle of minimizing the expected loss under given loss function. It incorporates both model uncertainty—through operating over a diverse pool of candidates—and a reference-based evaluation metric, which brings complementary signal about translation equivalence. MBR scoring starts with a set of candidate translations $\mathcal{H}$ sampled from a model, and then uses a reference-based metric $u$ to estimate the expected loss with respect to the other candidates, i.e.,

$$y^{\text{mbr}} = \text{argmax}_{h \in \mathcal{H}} \frac{1}{|\mathcal{H}|} \sum_{r \in \mathcal{H}} u(h, r) \tag{2}$$

This has a time complexity quadratic in $|\mathcal{H}|$, arising from the nested $\max$ and $\sum$ operations.

The MBR technique incorporates several plausible translations from the model, not just the mode. This feature, as well as the ability to integrate strong reference-based metrics, leads to the substantial gains in output quality over beam search. Previous research (Finkelstein et al., 2023) has shown that MBR decoding can be used with knowledge distillation, by using $y^{\text{mbr}}$ to replace $\hat{y}$ in Eq 1, resulting in improvements over beam search baselines.

**MBR-$n$** Inspired by these impressive results, we ask whether other candidates that received high MBR scores are also of high quality, and useful in distillation. To test this idea, we use for supervision the set $\mathcal{Y}^{\text{mbr}}$, comprising the top $n$ scoring candidates from $\mathcal{H}$ (computed as the $n$ max solutions to Eq. 2). This gives the distillation loss,

$$\mathcal{L}_{\text{MBR-}n} = -\frac{1}{|\mathcal{Y}^{\text{mbr}}|} \sum_{y \in \mathcal{Y}^{\text{mbr}}} \log p(t = y|s) \tag{3}$$

In summary, our method ('MBR-$n$' hereafter) works as follows: given a teacher model and a set of monolingual input sentences, we generate a set of candidate translation samples for each input sentence. Next, we conduct MBR scoring on the candidate set and select the top $n$ scoring candidates to form a training set. Finally, we train the student model using supervised training on this dataset. Letting the student model see more high-quality outputs from teacher can help them better approximate the teacher's distribution. The workflow of MBR-$n$ shown in Figure 1.

## 3 EXPERIMENTS

In this section, we will introduce the translation language, datasets, models and evaluation.

**Languages and dataset** We conduct experiments on two language pairs: English to German (en-de) and English to Japanese (en-ja), following Finkelstein et al. (2023), where en-de represents a high-resource language pair and en-ja a medium-resource language pair.

We used two types of training data. The first is **base finetuning data**, which primarily serves to train teacher models, ensuring satisfactory translation performance. Additionally, we used this dataset to fine-tune the student model, observing how our method enhances the performance of a student model already proficient in translation. We employ the WMT22 en-de and en-jp training sets for this purpose, which consist of about 20M and 8M sentence pairs, respectively. The second type of data is **KD data**, employed for SeqKD training. As the teacher's outputs are used, only monolingual input text is required, however to allow for benchmarking against human-annotated reference training, we use a high-quality parallel corpus, the source sentences serve as **KD data**, while the target sentences are exclusively used for the Reference-based training baseline. Following (Finkelstein et al., 2023), for the KD dataset we use the aggregated test sets from WMT09-19 (en-de) and the WMT21 test set for en-ja, which contain ~30k and ~1k instances, respectively.

**Models** We conducted experiments using various sizes of student and teacher models to assess different methods across different scales. We use PaLM2 (Anil et al., 2023), a general purpose multilingual large language model capable of excellent translation performance. For student models, we consider two of the smallest PaLM2 models: XXXS and XXS; while for the teacher we use XXS, XS, S and L[1] model sizes. These models are referred to as Gecko (XXS), Ot-

---

[1]Only used for en-de translation, not for en-jp.

ter (XS), Bison (S) and Unicorn (L) on Google Cloud. The smallest XXXS model is a smaller transformer designed to have size comparable to the big transformer in Vaswani et al. (2017).

We present the performance of the teacher model in Table 1. Teacher models were fine-tuned separately for each language pair, with **base fine-tuning dataset**. Exceptions include: the en-de L model, which was not fine-tuned due to its pre-existing strong performance; and the en-de XS and S models, which were trained using SeqKD with MBR supervision against the L model and in-house data. Note that the scores here for the XXS model differ to 'Reference' training in Tables 2 and 3.

Table 1: BLEURT scores for teacher models, evaluated on the respective WMT22 test set.

| Lang. pair | XXS | XS | S | L |
|---|---|---|---|---|
| en-de | 0.7552 | 0.8008 | 0.8034 | 0.7973 |
| en-jp | 0.6678 | 0.6857 | 0.7156 | - |

**Baseline**    As baselines we include **Ref**erence based training, i.e., using the human translated ground truth data; **Beam** search outputs from the teacher model, the standard SeqKD approach (Kim & Rush, 2016); **Rand**om sampled outputs from the teacher model;[2] and the **MBR** sequence computed using 256 teacher samples (computed as above) and the BLEURT (Sellam et al., 2020) metric(Finkelstein et al., 2023), equivalent to MBR-$n$ with $n = 1$. We use the same configuration for our method, **MBR-**$n$, but simply take the top $N$ sequences for KD training.

**Generation**    We use the teacher model to generate target side outputs for source sentences, using a zero-shot prompt specifying the source and target language. E.g.,

> *English:* `source sentence`
> *German:*

We use epsilon sampling (Hewitt et al., 2022) for teacher models with $\epsilon = 0.02$ to generate translation candidates for each source sentence. For MBR-$n$, we choose 256 to be the number of candidates and employ BLEURT (Sellam et al., 2020) as the metric function to calculate MBR scores for each candidate.For model evaluation, we generate outputs with greedy decoding.

**Evaluation**    The validation set for en-de is the WMT21 test set, while for en-jp, we use the WMT22 dev set. We primarily use the WMT22 testset to test the performance of student models, with BLEURT. We confirm also our key findings hold for out-of-domain generalization using the Flores dataset (Guzmán et al., 2019), and with different metrics, namely sacreBleu (Post, 2018), ChrF (Popović, 2015) and Comet22 (Rei et al., 2022).

## 4 RESULTS AND DISCUSSION

In this section, we present our main experimental results, compare the outcomes of MBR-$n$ with other baselines, report the performance for en-de and en-jp language pairs, and discuss the capacity curse and data efficiency.

**KD better than Reference-based training**    Figure 2 compares baseline reference-based training (left column, Ref) to several knowledge distillation methods. All knowledge distillation methods outperform directly learning from human references. Notably, using MBR to generate teacher outputs yields much better results than using beam search. This finding holds across many other experimental settings (see Table 2), and is consistent with the findings of Finkelstein et al. (2023).

**MRB-**$n$ **outperforms MBR**    MBR-$n$ results in further consistent gains across different model sizes and language pairs, as shown in Table 2. Our approach yields the greatest improvements for smaller student models, with performance steadily increasing as $N$ grows. In most settings the optimal result is achieved at the highest value of N=40. An outlier is en-de with the PaLM2-XXS student, where

---

[2]Samples are drawn using epsilon sampling (Hewitt et al., 2022) with $\epsilon = 0.02$. We also tested candidate selection with temperature sampling, see details in Appendix A.

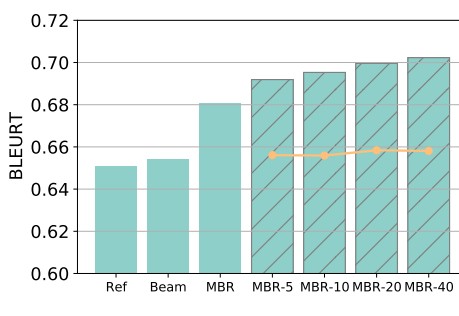 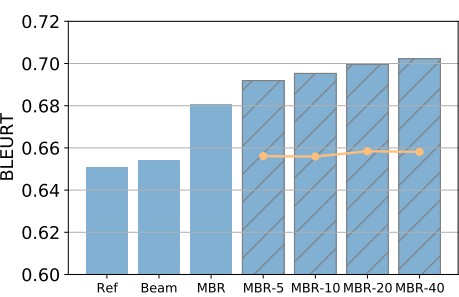

(a) XXXS student. Initial BLEURT 0.2537.   (b) XXXS-FT student. Initial BLEURT 0.6810.

Figure 2: Comparing a pre-trained student (a) versus one fine-tuned for translation (b). Here a XXXS student is trained against a XXS teacher on en-de. Reported in the caption are the BLEURT scores for the student models before KD training; the accuracy of the teacher is 0.7552, as reported in Table 1. The yellow line shows the effect of training on $5 \ldots 40$ random samples.

Table 2: Translation results for English to German (Top) and English to Japanese (Bottom) translation comparing standard 'reference' based fine-tuning against KD training with various teacher decoding strategies, evaluated using BLEURT on the WMT22 test set. These results are below those of the teacher models, see Table 1.

| | **Student** | **XXXS** | | | | **XXS** | | |
|---|---|---|---|---|---|---|---|---|
| | **Teacher** | **XXS** | **XS** | **S** | **L** | **XS** | **S** | **L** |
| en→de | Reference | 0.6511 | | | | 0.7630 | | |
| | Beam | 0.6542 | 0.6629 | 0.6632 | 0.6521 | 0.7760 | 0.7753 | 0.7662 |
| | MBR | 0.6806 | 0.6778 | 0.6744 | 0.6691 | 0.7837 | 0.7843 | 0.7796 |
| | MBR-5 | 0.6919 | 0.6837 | 0.6823 | 0.6744 | 0.7874 | 0.7857 | 0.7804 |
| | MBR-10 | 0.6953 | 0.6862 | 0.6812 | 0.6750 | 0.7877 | 0.7852 | 0.7791 |
| | MBR-20 | 0.6996 | 0.6865 | 0.6833 | 0.6767 | 0.7877 | 0.7854 | 0.7796 |
| | MBR-40 | 0.7023 | 0.6889 | 0.6855 | 0.6762 | 0.7873 | 0.7861 | 0.7780 |
| en→jp | Reference | 0.5158 | | | | 0.6679 | | |
| | Beam | 0.5083 | 0.5549 | 0.5351 | - | 0.6671 | 0.6748 | - |
| | MBR | 0.5344 | 0.5314 | 0.5356 | - | 0.6747 | 0.6770 | - |
| | MBR-5 | 0.5570 | 0.5542 | 0.5514 | - | 0.6827 | 0.6852 | - |
| | MBR-10 | 0.5722 | 0.5628 | 0.5575 | - | 0.6841 | 0.6878 | - |
| | MBR-20 | 0.5733 | 0.5657 | 0.5638 | - | 0.6844 | 0.6884 | - |
| | MBR-40 | 0.5754 | 0.5701 | 0.5649 | - | 0.6845 | 0.6893 | - |

performance peaks at MBR-5, beyond which the performance of some models declines slightly. This trend may be attributed to the fact that large student models have already achieved performance levels close to that of the teacher model (PaLM2-XS: 0.8008), making further enhancements challenging as the student approaches the performance ceiling.

**MBR scoring is important**   A key question is whether the effectiveness of MBR-$n$ distillation comes purely from the use of several outputs for each input sentence (and thus an effectively larger KD training dataset), versus the MBR top-$n$ selection method. To test this hypothesis we compare against random selection (Rand in Figure 2). Using random samples from the teacher lead to inferior performance, roughly at the level of Ref and Beam baselines.

**Strong students**   We used **base finetuning data** to finetune en-de PaLM-XXXS and PaLM-XXS model, denoted XXXS-FT and XXS-FT respectively. This process aimed to create student models already proficient in translation, allowing us to assess whether our method could enhance a student model with strong translation ability. We confirm that our distillation method also works well with stronger student models. As presented in Table 3 and Figure 2b, we observed the same overall conclusions, albeit with slightly higher BLEURT scores for baselines and distillation methods.

Table 3: BLEURT scores for English to German translation using KD training, evaluated on the WMT22 test set. Student models are fine-tuned on large supervised datasets before distillation training. This contrasts with Table 2, which reports results for pre-trained student models. There is still a performance gap for the best models of at least 0.01 to the teacher scores, as listed in Table 1.

| Student | XXXS-FT | | | | XXS-FT | | |
|---|---|---|---|---|---|---|---|
| Teacher | XXS | XS | S | L | XS | S | L |
| Reference | | 0.6511 | | | | 0.7630 | |
| Beam | 0.6877 | 0.6968 | 0.6927 | 0.6864 | 0.7802 | 0.7840 | 0.7722 |
| MBR | 0.7050 | 0.7047 | 0.7034 | 0.6990 | 0.7896 | 0.7887 | 0.7845 |
| MBR-5 | 0.7094 | 0.7059 | 0.7026 | 0.6982 | 0.7902 | 0.7887 | 0.7841 |
| MBR-10 | 0.7116 | 0.7072 | 0.7028 | 0.6990 | 0.7898 | 0.7890 | 0.7834 |
| MBR-20 | 0.7122 | 0.7079 | 0.7036 | 0.6993 | 0.7893 | 0.7900 | 0.7833 |
| MBR-40 | 0.7143 | 0.7070 | 0.7049 | 0.6981 | 0.7903 | 0.7895 | 0.7812 |

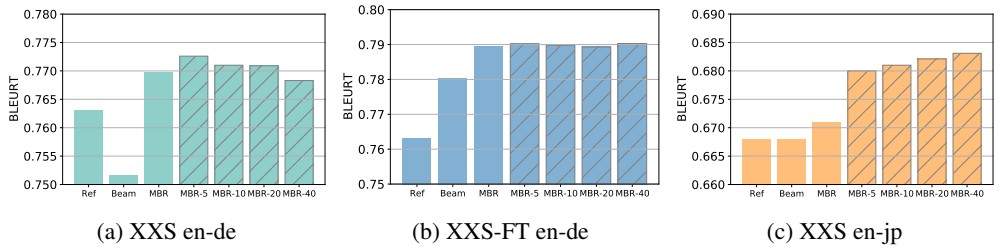

(a) XXS en-de     (b) XXS-FT en-de     (c) XXS en-jp

Figure 3: Self training for XXS model, models were trained with its own output.

**Teacher performance** When compared teacher performance (shown in Talbe 1 to the student model's performance shown in Table 2 and Table 3, it is evident that some student models can achieve results close to those of the teacher model. For instance, the XXS-FT student trained with MBR-40 from the teacher XS achieves a BLEURT score of 0.7903, leaving only a small gap from the teacher's performance of 0.8008.

**Self-training** Figure 3 presents the results of self-training experiments with the XXS model, a process akin to knowledge distillation but without a separate teacher model but using the student model's own output for further training. The results demonstrate that the MBR-$n$ method is also effective in a self-training context. MBR-based self-training yielded better performance than reference training for the XXS en-de model, while beam search self-training resulted in lower scores. Furthermore, increasing the $n$ improved self-training effectiveness for en-jp.

**Diversity** MBR-$n$ generates a wide range of translation candidates for each source input, expanding the spectrum of potential translations seen by the student model. Consequently, we evaluate the

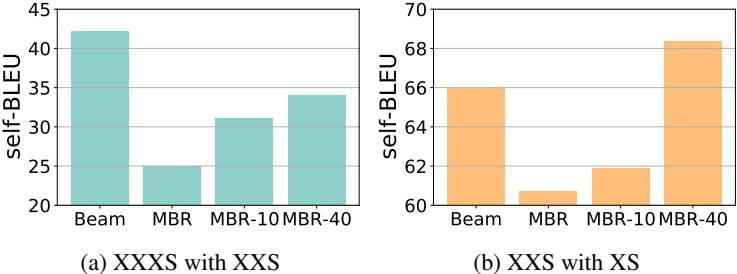

(a) XXXS with XXS     (b) XXS with XS

Figure 4: Diversity of outputs measured using self-bleu. Two settings are illustrated: student PaLM2-XXXS trained with teacher PaLM2-XXS and student PaLM2-XXS trained with teacher PaLM2-XS, on English to German translation task.

uncertainty of the student model using self-BLEU metrics to determine whether the distilled student also exhibits high output diversity. We use epsilon sampling method to generate 5 outputs for each input from the student model. Subsequently, we compute the sacreBLEU score for each pair of outputs and average them. A higher self-BLEU score indicates greater gram overlap in the generated outputs, signifying lower diversity. Conversely, a lower self-BLEU score signifies higher diversity. We assess self-BLEU on the WMT22 test set, for en-de translation.

Figure 4 reveals that Beam distilled students exhibit relatively low diversity, evidenced by high self-BLEU. This outcome aligns with our hypothesis, as each instances has a single output and beam search outputs tend to use consistent translation decisions. Concerning the MBR-$n$ method, intuitively, it introduces greater diversity and uncertainty to the student model through training. However, we were surprised to observe that this does not map to output uncertainty: MBR distilled models display the most uncertainty, despite each input having only one corresponding translation. As $n$ increases in MBR-$n$, we noticed a decrease in the distilled student's output diversity. Why this happens is an open question, that requires further exploration. Note that unlike Beam, MBR-40 distillation also enhances model performance substantially.

**Capacity curse** The results from Table 2 and Table 3 also illustrates the *capacity gap curse* phenomenon. Essentially, employing a better and larger teacher model to instruct the student model doesn't always lead to better student performance; instead, performance degrades. This curse can be seen for our MBR-$n$ method, and most other distillation strategies. Notably the PaLM2-XXS teacher has the lowest performance on its own, however it produces some of the best results for student distillation (e.g., with student XXXS, for both en-de and en-jp). This phenomenon has been extensively discussed in prior literature (Zhang et al., 2023a). Our experiments, conducted on a broader scale, validate that this issue persists for translation distillation with LLMs.

**Staged training** Given the capacity curse, we propose a curriculum learning approach for training (Bengio et al., 2009), which we call *staged training*. This works by first training the student with weak teacher, then continuing training with a strong teacher. The idea is that once the student is capable at translation, it will be better able to learn the deeper nuances from the larger teacher. We test this idea with PaLM2-XXXS with teachers PaLM2-XXS and PaLM2-XS.

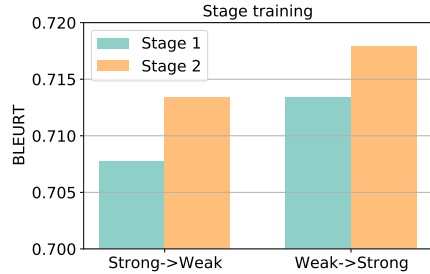

Figure 5: Staged training for en-de, where student is trained in a two stage curriculum against different teachers.

For comparison, we also run the process in reverse: first using the strong teacher, followed by the weaker teacher. The results are illustrated in Figure 5. Through the Weak→Strong approach, we observed that this method indeed leads to additional improvement in the student model's performance (from 0.7134 to 0.7179). Conversely, with the Strong→Weak approach, the performance only matches that achieved when using weak teacher alone to train student (0.7134, identical to stage 1 of the weak→strong experiment). Nevertheless, upon employing this training method with the larger PaLM2-XXS student, we observed no improvements. This can be attributed to the fact that PaLM2-XXS students already operate near the upper threshold of the teacher model.

**Runtimes** Table 4 show the gradient update steps for the best en-de student model checkpoint trained under various teacher supervision settings. The training time for MBR-$n$ increases linearly with the value of $n$. However, it is notable that MBR-$n$ can achieve similar or better results compared to other methods within the same number of update steps, although it requires more time to converge to the optimal value.

**Data efficiency** As demonstrated in the en-jp experiments, MBR-$n$ exhibits strong performance even with limited data (1K samples used in KD training). Motivated by this, we conducted experiments using the en-de dataset, by sub-sampling the KD dataset to explore the impact of dataset size in KD. Subsequently, we trained student models using different MBR-$n$ approaches, and the results are shown in Figure 6. Our findings indicate that with highest $N = 40$, the models are

Table 4: Runtime comparison showing training cost for knowledge distillation training with different teacher supervision settings, for en-de translation. The Beam column shows the number of gradient update steps until the stopping condition is reached for the Beam supervision, while the columns to the right show the relative number of identically sized steps required for other supervision methods. Observe that while MBR-$n$ expands the training set beyond Beam and MBR by a factor of $n$, the training cost sub-linear in $n$.

| Student | Teacher | Beam | MBR | MBR-5 | MBR-10 | MBR-20 | MBR-40 |
|---------|---------|------|-----|-------|--------|--------|--------|
|         | XXS     | 12,500 | 0.80 | 1.88 | 1.76 | 4.16 | 6.92 |
| XXXS    | XS      | 8,000  | 1.38 | 1.88 | 1.75 | 1.75 | 3.06 |
|         | S       | 9,500  | 0.89 | 1.21 | 1.32 | 1.63 | 1.68 |
|         | L       | 7,500  | 1.33 | 1.40 | 1.47 | 1.87 | 2.07 |
|         | XS      | 10,000 | 1.05 | 2.70 | 4.05 | 5.75 | 6.20 |
| XXS     | S       | 11,500 | 1.13 | 1.61 | 1.57 | 2.22 | 7.83 |
|         | L       | 9,500  | 1.53 | 2.16 | 2.11 | 2.16 | 4.42 |

Figure 6: MBR-$n$ is more data efficient than baseline methods, in terms of the volume of distillation training data required. Shown above are results for English to German translation with two student/teacher configurations. KD instances are measured in thousands of sentences, with the rightmost 30k setting corresponding to the complete KD dataset.

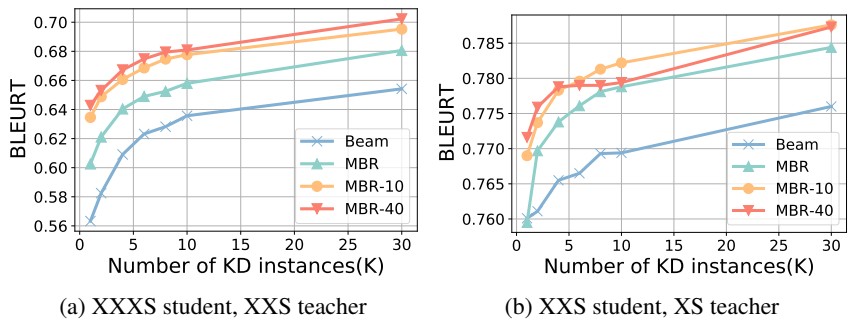

(a) XXXS student, XXS teacher          (b) XXS student, XS teacher

about 3x more data efficient than MBR, and about 15x more data efficient than beam. Here the input data is relatively cheap (quality monolingual text), suggesting large distillation sets should be used. Another consideration is compute-efficiency: once the expensive step of the MBR scoring is completed (quadratic in candidate samples, linear in dataset size) the difference in training cost for MBR-$n$ vs MBR-1 is modest ($1.2\times$ to $7.8\times$). Overall, MBR-$n$ can improve accuracy in both data-scarce and data-rich scenarios, at small cost.

**Overfitting to BLEURT** Given we use BLEURT as the metric function for MBR scoring, there are concerns about the model's remarkable performance in the BLEURT test being potentially linked to overfitting with BLEURT. Thus, we showcase the model's performance across various other evaluation metrics in Table 5, encompassing sacreBLEU (Post, 2018), chrF (Popović, 2015), and COMET22 (Rei et al., 2022). The results reveal that MBR-$n$ consistently exhibits superior performance of the student model across a range of evaluation metrics. These outcomes indicate that our approach is not overfitting to BLEURT, but rather the same findings hold of other evaluation methods.

**Out of domain evaluation** To test whether our results generalize to other domains, we evaluated the performance of our en-de approach using an out-of-domain (OOD) dataset, Flores, as illustrated in Table 5. This test set is derived from Wikipedia, versus WMT22 which is drawn from news media, and closely matches the training and KD datasets. The results demonstrate that the MBR-$n$ approach performs well for both in domain and out of domain evaluation. MBR-40 consistently achieved the highest scores across all evaluation metrics on the Flores test set. This suggests that the improvements facilitated by MBR-$n$ are comprehensive, extending beyond merely in-domain data. By exposing

Table 5: Findings carry over to other evaluation domains and metrics. Showing XXXS student for English to German with XXS teacher, evaluated over WMT22 development set and Flores test set. The best result for each metric is highlighted in **bold**. The version of **BLEU** is **sacreBLEU**.

| | WMT22-dev | | | | Flores | | | |
|---|---|---|---|---|---|---|---|---|
| | **BLEU** | **chrF** | **BLEURT** | **COMET22** | **BLEU** | **chrF** | **BLEURT** | **COMET22** |
| Beam | 19.51 | 49.13 | 0.5991 | 0.7394 | 25.35 | 54.88 | 0.6619 | 0.7796 |
| MBR | 16.20 | 46.89 | 0.6254 | 0.7432 | 22.09 | 52.38 | 0.6830 | 0.7928 |
| MBR-5 | 19.49 | 50.08 | 0.6469 | 0.7709 | 25.02 | 53.93 | 0.6963 | 0.8038 |
| MBR-10 | 18.23 | 49.44 | 0.6441 | 0.7686 | 25.20 | 54.04 | 0.6999 | 0.8087 |
| MBR-20 | 19.52 | 50.38 | 0.6516 | 0.7797 | 25.88 | 54.45 | 0.7048 | 0.8103 |
| MBR-40 | **20.94** | **51.24** | **0.6555** | **0.7844** | **26.85** | **55.91** | **0.7117** | **0.8178** |

the student model to a broader spectrum of translation scenarios, it can better assimilate various translation methods and exhibit enhanced performance in dealing with OOD data.

## 5 RELATED WORKS

**Minimum Bayes risk**  Minimum Bayes Risk (MBR) decoding, originating from statistical decision theory, aims to minimize the expected risk by selecting the decision that minimizes the expected loss over all possible decisions. It outperform MAP beam search in various tasks (Suzgun et al., 2023; Shi et al., 2022), including machine translation. Initially explored in statistical machine translation (Kumar & Byrne, 2004), MBR has recently garnered attention in NMT (Eikema & Aziz, 2022; Freitag et al., 2022) as a method to mitigate biases inherent in MAP decoding techniques such as beam search. However, MBR comes with a significant computational cost, necessitating the generation of numerous samples and the computation of utility metric functions quadratically proportional to the number of samples. Thus, efforts have been made to mitigate this computational burden, with some studies (Cheng & Vlachos, 2023) focusing on reducing the resource consumption of MBR. Additionally, besides using MBR for decoding during the inference stage, there have been efforts to integrate MBR into the training process. Shen et al. (2016) introduced MBR training, which involves training with evaluation metrics, while Finkelstein et al. (2023) proposed using the outputs of MBR decoding instead of beam search for KD training of the student model.

**Knowledge distillation**  Knowledge distillation (KD) (Hinton et al., 2015) is a model compression technique that employ a teacher model to guide the training of the student model, aiming to procure a smaller model that closely mirrors the teacher's behaviours and performance. Kim & Rush (2016) introduced the first KD technology for machine translation, known as sequence-level knowledge distillation (Seq-KD). This straightforward approach, training the student model with text generated by the teacher, has been widely employed in machine translation (Tan et al., 2019; Wang et al., 2021; Zhang et al., 2023b) and the other generation tasks. With the remarkable success of LLMs, many studies have begun employing KD techniques on LLMs. Some approaches (Peng et al., 2023; Zhou et al., 2023; Finkelstein et al., 2023) follow Seq-KD and fine-tune LLMs using teacher-generated text. Gu et al. (2023) and Agarwal et al. (2024) leverage imitation learning and frame distillation, framing the problem as a reinforcement learning (RL) task. They both replace forward KL divergence with reverse KL divergence, as it better suits generation tasks. Finkelstein et al. (2023) present a method similar to ours, employing MBR and Quality Estimation to rank candidates for distillation. However, their approach stops at selecting the best candidate, while our method explores the impact of multiple candidates and conducts a deeper analysis of multiple inputs.

## 6 CONCLUSION

We present a novel approach to KD of LLMs, MBR-$n$. By leveraging multiple outputs of MBR scoring, we train the student model to more effectively capture the teacher's distribution, resulting in improved performance. Our extensive experimentation spans en-de and en-jp translation tasks, encompassing diverse student and teacher model configurations. The findings underscore the efficacy of MBR-$n$.

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

## A  CANDIDATE SELECTION BY TEMPERATURE SAMPLING

MBR-$n$ selects sentences with the minimum Bayes Risk scores from the candidates. Here we try another technique, based on temperature sampling of outputs according to their Bayes Risk scores. When the temperature $t$ is small, the sampling is close to maximisation, i.e., our proposed MBR-$n$ approach, while larger values of $t$ approach the Random baseline. The results are depicted in Figure 7. We observe that smaller values of $t$ correspond to better student performance, supporting our technique of top-N MBR scoring.

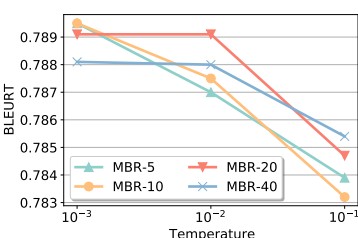

Figure 7: Candidate selection by temperature sampling, de-en translation.

