# OpenReview forum: "Don’t Throw Away Data: Better Sequence Knowledge Distillation"
_ICLR.cc/2025/Conference — Submitted to ICLR 2025_

### Official Review · Reviewer_Kaxk · 2024-10-31

**Soundness:** 3
**Presentation:** 3
**Contribution:** 2
**Rating:** 5
**Confidence:** 4

**Summary:**

This paper introduces MBR-n, a novel approach to enhance sequence-level knowledge distillation (SeqKD) for translation tasks. The approach extends the Minimum Bayes Risk (MBR) decoding methodology by utilizing multiple high-quality teacher outputs instead of a single output for distillation training. The authors demonstrate that this method improves student model performance in machine translation across different language pairs and model sizes, particularly when using high-capacity teacher models.

**Strengths:**

1. MBR-n’s use of multiple high-MBR-scoring candidates for distillation presents a more comprehensive knowledge transfer from teacher to student.
2. Simple and effective method with comprehensive analysis.

**Weaknesses:**

1. Line 84, I'm confused by "confirm the capacity gap curse is an open issue", what do the authors mean by "open issue"
2. Figure 2, by `beam`, does the authors means using the top1 output as the KD target generated by beam search? Do the authors try like Beam-n methods?
3. While MBR-n outperforms MBR-1, the computational cost is also n times, whether teacher MBR generation and KD process. Even though the authors claim that MBR-n is more data efficient, the experiments are mostly on small size of data. Larger orders of magnitude of the data scale would make the results more convincing.
4. More experiments on more language pairs would make the results more convincing
5. Other than BLEU, BLEURT, better translation evaluation metrics, e.g, COMET should be taken into consideration [1]

[1] Tom Kocmi, Vilém Zouhar, Christian Federmann, and Matt Post. 2024. Navigating the Metrics Maze: Reconciling Score Magnitudes and Accuracies. In Proceedings of the 62nd Annual Meeting of the Association for Computational Linguistics (Volume 1: Long Papers), pages 1999–2014, Bangkok, Thailand. Association for Computational Linguistics.

**Questions:**

Please see weaknesses

---

> ### Author Response · Authors · 2024-11-21
>
> Thank you for your review. Below are our responses, and we hope they address your concerns.
>
> 1. The open issue means this issue has been discussed in many papers[1], yet a satisfactory solution has not been found.
>
> 2. Yes, we used the top-1 output with beam size = 4 as the KD target. We didn’t experiment with beam-n methods because, as suggested in the Seq-KD paper. We observed that beam-search outputs are often too similar, so this would provide very little learning signal.
>
> 3.  The computation of MBR-N is not simply N times the cost of MBR-1. For MBR-1, a large number of outputs still need to be generated, and MBR scoring must be performed. Our method, on the other hand, considers more candidates instead of just selecting the top-1, so the time required to generate KD data remains exactly the same. Regarding training time, here we report the effect of different n values under the same update step. For the En-De task, we tested XXS student distilled from XS teacher and XXXS student distilled from XXS teacher, using 10k instances and 10k training steps. The results indicate that our method surpasses the baseline MBR, further demonstrating its runtime efficiency, with the same number of training update steps, MBR-N achieves better performance than MBR-1.
>
> |Model | MBR | MBR-10 | MBR-40 |
> | --------| -------- | ------- | ------- |
> | XS - XXS | 0.7781| 0.7807 |0.7797|
> |XXS-XXXS | 0.6407 | 0.6711 |0.6760|
>
> The KD data we used is not large, but this is a reasonable setting [2]. For KD, a large dataset is not necessary. In particular, for LLMs, training with a small amount of KD data can already yield excellent results.
>
> 4. We acknowledge that experimenting with additional language pairs would enhance the results; however, due to limitations in computational resources, we have not yet explored this approach. The selection of language pairs and instances follows the setup described in [2]. Furthermore, our current results already demonstrate the significant advantages of our approach.
>
> 5. Thanks for the suggestion, we have included COMET22 results in Table 5. This is important to mitigate the risk of overfitting to the metric used in MBR (Bleurt) as part of our training as well as evaluation. We observe the same overall findings using COMET as Bleurt, confirming the utility of our method.
>
>
> [1] Zhang, Chen, et al. "Lifting the curse of capacity gap in distilling language models." ACL 2023.
>
> [2] Finkelstein, Mara, et al. "Mbr and qe finetuning: Training-time distillation of the best and most expensive decoding methods.", ICLR 2024.

---

> > ### Comment · Reviewer_Kaxk · 2024-11-26
> >
> > Thank you for your response.
> > 1. About `open issue`. Please do use clearer and more precise definitions in future drafts.
> > 2. Did the authors try to use sampling methods + beam search, to achieve more diverse outputs?
> > 3. MBR-40 is not better than MBR-10 and MBR, in the 1st line. I'm not convinced by the authors' arguments.
> > 4. `For KD, a large dataset is not necessary. `. I have to say I do NOT agree with this argument. Researchers noticed a long time ago about the KD small-data pitfalls [1]. Data size does matter for KD. I would appreciate if the authors do try larger scaled training for more convincing results.
> >
> > As a result, I decide not to change the score.
> >
> > [1] Hao, Z., Guo, J., Han, K., Hu, H., Xu, C., & Wang, Y. (2023). VanillaKD: Revisit the Power of Vanilla Knowledge Distillation from Small Scale to Large Scale. ArXiv. https://arxiv.org/abs/2305.15781

---

### Official Review · Reviewer_WsHB · 2024-11-03

**Soundness:** 3
**Presentation:** 3
**Contribution:** 2
**Rating:** 5
**Confidence:** 4

**Summary:**

This paper introduces a novel technique for knowledge distillation (KD) of large language models (LLMs), named MBR-n. By utilizing multiple outputs scored by MBR, the student model could better capture the teacher's distribution, leading to enhanced performance. The extensive experiments cover en-de and en-jp translation tasks, involving various configurations of student and teacher models, showing the effectiveness of MBR-n on the KD.

**Strengths:**

This paper proposes an effective knowledge distillation method to enhance the translation capabilities of large language models with relatively small sizes, outperforming traditional sequence-level knowledge distillation methods based on beamsaerch.

**Weaknesses:**

1. The MBR method was proposed a long time ago, and the author merely uses MBR to generate candidates for the Teacher model, lacking a certain degree of innovation.

2. The author's introduction to the MBR method is not detailed enough, which may confuse readers.

3. Most of the experimental results are compared with beam search-based baselines, lacking comparisons with more KD (Knowledge Distillation)-related work, such as https://aclanthology.org/2023.emnlp-main.178/

**Questions:**

1. Regarding the experimental results in Table 2, why is it that the larger the size of the Teacher model, the worse the translation quality (i.e., the lower the BLEURT score) of the distilled Student model? Please provide a detailed explanation.

2. Why did the authors choose PaLM as the baseline model? Did you ever try using the LLaMA language model or the BLOOM multilingual large model?

3. Why was zero-shot used when generating translation candidates using the Teacher model? Have you compared the impact on the Student model's performance when using the few-shot method to generate translation candidates?

4. Does using MBR to generate translation candidates for knowledge distillation have a significant impact on training efficiency? Would like to see a comparison of the training efficiency among Reference, Beam, and MBR.

5. Intuitively, the method proposed in this paper mainly aims to enhance the diversity of translations, right? Besides, are the references in the test set single-reference or multi-reference during the evaluation? If the evaluation is based on multi-reference, it is reasonable for the translation quality to improve. However, if it is based on single-reference, I think there should not be a significant improvement in translation quality, right?

---

> ### Author Response · Authors · 2024-11-21
>
> Thank you very much for your suggestions for our article. We hope to answer some of your questions here.
>
> 1. For novelty, we agree that the idea is straightforward. However, we show that the method is also highly effective. Added to this, we explored in depth topics such as data efficiency and the curse of dimensionality. It was previously widely believed that generating multiple distinct references for a source sentence would harm model performance. However, our method has demonstrated that this approach can actually improve the model's performance on LLM.
>
> 2. Thank you for pointing that out. We will revisit the Method Section and clarify the issue. We would greatly appreciate it if you could specify which parts are unclear, as that would be very helpful.
>
> 3. Thank you for pointing out the missing important citation. We will add it to the related work section, as it is relevant. However, this paper focuses on the question of how to train multiple students together, whereas our aim is to train a single, stronger student. Since the research questions are different, a direct comparison is challenging.
>
> ** Here are the responses to your questions below: **
>
> Q1: This is because the largest model was not fine-tuned on the specific translation data. However, it already demonstrates strong performance, and fine-tuning such a large LLM requires significant computational resources, which is why we did not fine-tune it.
>
> Q2: The choice of PaLM follows [1]. We agree that testing with more LLMs would lead to a stronger conclusion. However, due to time and computational resource limitations, we were unable to conduct such experiments.
>
> Q3: Since our focus is on testing KD rather than prompt learning, we opted for zero-shot, as it already produces good translation results. Therefore, we did not experiment with different shot configurations for generation.
>
> Q4: We have reported the runtime in Table 6, which shows the time required to achieve optimal performance. For the En-De task, we tested XXS student distilled from XS teacher and XXXS student distilled from XXS teacher, using 10k instances and 10k training steps. The table below demonstrates that, within the same number of training steps, our method outperforms the baseline MBR. Additionally, MBR decoding alone offers a significant improvement over greedy/beam/gold standard training, justifying the more resource-intensive algorithm. In contrast, our extension is relatively lightweight. Due to time constraints, we have not run experiments with Beam/Reference, but we will include them in the revision if the paper is accepted.
>
> |Model | MBR | MBR-10 | MBR-40 |
> | --------| -------- | ------- | ------- |
> | XS - XXS | 0.7781| 0.7807 |0.7797|
> |XXS-XXXS | 0.6407 | 0.6711 |0.6760|
>
> 5. The test set is single-reference, and the improvements are substantial. It was previously believed that adding diversity on the target side would not enhance single-reference performance, but our paper demonstrates that for LLMs, this diversity can indeed lead to better results. We also analyze the diversity of student models trained with MBR-n in the Diversity Section, and find that a greater number of diverse references actually results in less diversity in the generated outputs. This insight is an important innovation and contribution of our paper.
>
> [1] Finkelstein, Mara, et al. "Mbr and qe finetuning: Training-time distillation of the best and most expensive decoding methods.", ICLR 2024.

---

> ### Comment · Reviewer_WsHB · 2024-12-01
>
> After reading the comments from other reviewers and the authors' responses, the authors answered most of my questions. But I am still concerned that the innovation of this work is not enough to appear in the conference, I will keep my score.

---

### Official Review · Reviewer_pQnb · 2024-11-03

**Soundness:** 2
**Presentation:** 3
**Contribution:** 1
**Rating:** 3
**Confidence:** 4

**Summary:**

The authors build on existing MBR methods to generate samples for distillation. While it is a well-known approach, the authors’ main discovery is that instead of using one MBR sample, using multiple MBR samples can be better.

**Strengths:**

1. The authors study knowledge distillation, which is an important research direction to make LLMs more accessible.
2. The authors have efficiency analysis, showing their approach is linearly slower in terms of number of samples used.

**Weaknesses:**

1. Lack of novelty. The authors propose to use multiple samples from MBR for distillation, which is essentially tuning a hyper-parameter of an existing approach rather than a novel method.
2. Little gain for the extra training. As shown in Table 2, even using 40 times as much training data, the performance can be as little as ~0.02 in the XXXS student setting. The performance can even degrade in many cases for the XXS student setting.
3. No analysis about the number of candidates required. The current approach is slow, not only because the authors require 40 times training samples; it is also slow because for each input, the authors generate 256 candidate samples for MBR to select from. This could be impractical for most scenarios.

**Questions:**

How similar are the top-n MBR samples?

---

> ### Author Response · Authors · 2024-11-21
>
> Thank you very much for your feedback, we would like to clarify a few points here.
> 1. For novelty, we agree that the idea is straightforward. However, we show that the method is also highly effective. Added to this, we explored in depth topics such as data efficiency and the curse of dimensionality. It was previously widely believed that generating multiple distinct references for a source sentence would harm model performance. However, our method has demonstrated that this approach can actually improve the model's performance on LLM.
>
> 2. Regarding the effectiveness of our method, we conducted the main experiments using BLEURT in the paper. A score of 0.02 represents a significant improvement. Specifically, we observed a +0.015 increase in BLEU, approximately +0.02 in chrF, and around +0.05 in COMET22, compared to the baseline beam search decoding for XXXS student distillate from XXS teacher, as shown in Table 5. These changes are considered substantial and meaningful, particularly in comparison to state-of-the-art models.
>
> 3. The "Data Efficiency" section discusses the number of candidates, with Figure 5 showing how varying candidate numbers affect performance. Additionally, generating multiple candidates is not highly resource-intensive, as it involves inference rather than backpropagation. Also, the different metrics can affect the cost of MBR, like QE would render it linear in complexity, not quadratic. In data-limited scenarios, we believe our approach can yield significant performance improvements. Table 6 showed the optimal training steps required to achieve the best performance. The table below shows that, within the same number of training steps, our method outperforms the baseline MBR. For the En-De task, we tested XXS student distilled from XS teacher and XXXS student distilled from XXS teacher, using 10k instances and 10k training steps. Moreover, MBR decoding alone provides a substantial improvement over greedy/beam/gold standard training, thus justifying the more expensive algorithm. In contrast, our extension is relatively lightweight.
>
> |Model | MBR | MBR-10 | MBR-40 |
> | --------| -------- | ------- | ------- |
> | XS - XXS | 0.7781| 0.7807 |0.7797|
> |XXS-XXXS | 0.6407 | 0.6711 |0.6760|
>
> **Here is the response to the question:**
>
> A: We extracted top-n MBR samples from the XS teacher model for 100 source instances, then calculated the average self-BLEU (by computing the BLEU score between different references for the same source sentence and averaging the results). The self-BLEU scores are as follows: MBR-5: 63.17, MBR-10: 62.28, MBR-20: 59.83, and MBR-40: 55.77. A higher self-BLEU indicates greater similarity among the references. As we increase the value of n, the similarity decreases. A self-BLEU around 60 suggests that the references are similar but not identical. We will include this discussion in our paper.

---

> ### Comment · Reviewer_pQnb · 2024-11-26
> **Response**
>
> Thank you for the response!
>
> I appreciate the new results, and I think they make the paper more comprehensive.
>
> However, I still share the concern regarding novelty with Reviewer WsHB and remain negative about this submission.

---

### Official Review · Reviewer_R6rU · 2024-11-04

**Soundness:** 3
**Presentation:** 2
**Contribution:** 2
**Rating:** 3
**Confidence:** 3

**Summary:**

The paper focuses on improving sequence knowledge distillation by extending minimum Bayes risk (MBR) decoding, calculating loss for top-n MBR decoded sequences instead of a single one. The authors conducted extensive experiments on translation tasks on high-resource (en-de) and medium-resource (en-jp) translation tasks, showing the improvement of their MBR-n approach over baseline methods in terms of model size scaling, data efficiency, showing the effectiveness of their proposed approach.

**Strengths:**

1. Simplicity: The paper proposes a simple and generalized method to improve existing knowledge distillation methods (MBR) by aggregating losses of multiple MBR decoded sequences.
2. Extensiveness of experiments: The authors conducted extensive experiments on en-de and en-jp translation tasks, providing convincing evidence that MBR-n improves over baseline methods and the vanilla MBR for the most part over various model, data, and training configurations.

**Weaknesses:**

1. Manuscript requires further polishing: While the writing of the manuscript is mostly coherent and easy to follow, there are several places that can use some further polishing, including but not limited to:

a) Multiple typos (such as "MRB-n outperforms MBR", "shown in Talbe 1" in section 4, etc.)

b) In section 4, the subsection titles are not uniformly formatting, that is, some titles are full sentences (MBR scoring is important); some adopt headline grammar (KD better than Reference-based training); and some do not summarize the finding of their subsections like the others (Teacher performance, Diversity, etc.)

2. Not enough novelty: The approach proposed in the paper seems more like a naive engineering increment upon the already existing MBR decoding approach. e.g. It would have been great if the authors had paid more attention to how the runtime convergence could be improved.

3. Confusing experimental setups: Teacher models (XXS, XS, S, L) are obtained in 3 different ways. This makes it hard to compare the effect of MBR-n in model scaling since it is not strictly one-factor-at-a-time.

4. Insufficient result interpretation:

a) The overall performance increment does not seem substantial, especially for models distilled from larger teacher models, or fine-tuned student models. Could have expanded on the Staged traininig experiments in section 4 and incorporated curriculum learning in the proposed approach to address the capacity curse.

b) In Section 4 Diversity, no explanation is made over why MBR-n student models have high diversity when n is small, and the diversity decreases when n is larger.

c) The method is expectedly not as efficient as the vanilla MBR in terms of runtime efficiency, but is more data efficient. Could have evaluated the overall efficiency between different methods during training by unifying the two aspects with a custom metric.

**Questions:**

1. Have you considered improving the runtime efficiency or addressing the capacity curse of the proposed methods on top of the naive approach of aggregating losses of multiple MBR sequences?

2. Could you provide experiments with teacher models obtained in the same way? Or can you explain the reason for this setup? For example, is it because that the performance margin for larger models between MBR-n and other approaches dimninishes even more when teacher models is stronger?

3. Could you explain the phenomenon where MBR-n with larger n gives lower diversity? Is there a better metric to evaluate the diversity of the outputs?

4. Could you provide overall efficiency evaluation unifying runtime efficiency and data efficiency and compare between baselines and your approach?

---

> ### Author Response · Authors · 2024-11-21
>
> We greatly appreciate your insightful comments. We will thoroughly review the paper, correct any typos, and refine the subsection titles to make them more concise and precise. First, we would like to clarify two points regarding novelty and effectiveness.
>
> 1. We agree that the idea is straightforward. However, we show that the method is also highly effective. Added to this, we explored in depth topics such as data efficiency and the curse of dimensionality. We agree that runtime convergence is an important and interesting direction for analysis, and our methods exhibit better runtime efficiency compared to the baseline. Table 6 highlights the impact of varying n on converge runtime. Here we will report the effect of different n values under the same update step. For the En-De task, we tested XXS student distilled from XS teacher and XXXS student distilled from XXS teacher, using 10k instances and 10k training steps. The results indicate that our method surpasses the baseline MBR, further demonstrating its runtime efficiency. This analysis and discussion will be included in the paper.
>
> |Model | MBR | MBR-10 | MBR-40 |
> | --------| -------- | ------- | ------- |
> | XS - XXS | 0.7781| 0.7807 |0.7797|
> |XXS-XXXS | 0.6407 | 0.6711 |0.6760|
>
>
> 2. According to the effectiveness. Our experimental results demonstrate the effectiveness of our method. In the distillation of the small model XXXS from XXS, we achieve a BLEURT improvement of 0.02 compared to beam search, this improvement is substantial, particularly when applied to state-of-the-art (SOTA) models. , Additionally, notable gains are also observed when using BLEU, chrF, and Coment22 metrics (see Table 5). While we acknowledge that the impact of distillation on the large model is less pronounced than on the smaller one, this is partly due to the "capacity curse" and partly because the distilled small model is already close to the observed performance limit, making further improvements challenging. This is an open question on how to do well. Employing curriculum learning to address the capacity curse is an interesting direction. Although it is not the main focus of this paper, we include the experimental results as a side exploration. We intend to further investigate Stage training as a potential solution to the capacity curse in future work.
>
> **Here are the responses to your questions below:**
>
> Q1: Thank you for the suggestion; it is indeed an interesting direction that we plan to explore in future work. Averaging the loss across tokens could potentially enhance training efficiency. Additionally, MBR with a reference-free metric (QE) has the potential to radically improve the runtime. It does however require very strong QE performance, which limits application to high resource settings. But this is the direction the field is headed, with auto raters etc.
>
> Q2: One reason for this is that our focus is on distillation from teacher models. The only requirement for the teacher models is that they must exhibit strong translation performance; we do not emphasize the specifics of their training process. We believe this does not impact the effectiveness of our method. Since training is resource-intensive, we prefer to reuse high-quality models and only fine-tune them if their performances are poor.
> That said, we agree that training the teacher model using the same method can help verify whether our approach is influenced by this factor. To test this, we fine-tuned the XS teacher model using the same method as the XXS teacher, using the WMT22 en-de data. The results show that our method still leads to significant improvement. The performance of the newly trained XS teacher is 0.7724, which is lower than the reported PaLM-XS performance of 0.8008 in the paper. The second row of below table corresponds to Table 2 in the paper, representing the En-De task with an XS teacher and an XXXS student.
>
> |Model | MBR | MBR-5 | MBR-10 | MBR-20 | MBR-40 |
> | --------| -------- | ------- | ------- | ------- | ------- |
> | XS - WMT | 0.6737| 0.6847 |0.6893| 0.6886 | 0.6864 |
> | XS |0.6778 |0.6837 |0.6862|0.6865|0.6889|
>
> Q3: As mentioned in the paper, this is an interesting open question. We have some hypotheses: it may be that adding more candidates improves learning of decoding distribution, as these candidates are of high quality. In contrast, with a single target, the output may be influenced more by the original LLM knowledge, leading to a more scattered distribution.
>
> Q4: We have discussed those in Table 4, Figure 5, and the "Runtime & Data Efficiency" section. We use training steps to assess runtime efficiency and evaluate data efficiency by testing with varying amounts of data. In the Table we show above(the first Table in point 1), our method showed the runtime efficiency is also better than vanilla MBR. If you have any additional questions, we would be happy to answer them, or if you have other metrics suggestions, we would gladly consider them.

---

> > ### Comment · Reviewer_R6rU · 2024-12-02
> >
> > Thank you for your response and I appreciate the additional results. Regarding Q4, I meant it would have been better if you had come up with a unified metric that could take both runtime efficiency and data efficiency into account.
> > But since I am still concerned about the novelty of the approach as well as the marginal gain from the baselines, especially if you increase n, I will not change my score as of now.

---

### Meta-Review · Area_Chair_QsVu · 2024-12-19

**Metareview:**

The paper proposes to use multiple MBR samples for knowledge distillaion.

While the proposed approach is simple and intuitive, reviewers raised several major concerns including the lack of novelty, insufficient analysis and interpretation of the results, as well as poor writing and formatting.

**Additional Comments On Reviewer Discussion:**

While the proposed approach is simple and intuitive, reviewers raised several major concerns including the lack of novelty, insufficient analysis and interpretation of the results, as well as poor writing and formatting. The reviewers unanimously recommended rejection.

---

### Decision · Program_Chairs · 2025-01-22

Reject